# Amino Acid Signaling for TOR in Eukaryotes: Sensors, Transducers, and a Sustainable Agricultural fuTORe

**DOI:** 10.3390/biom12030387

**Published:** 2022-03-02

**Authors:** Nanticha Lutt, Jacob O. Brunkard

**Affiliations:** 1Laboratory of Genetics, University of Wisconsin, Madison, WI 53706, USA; nantichalutt@berkeley.edu; 2Department of Plant and Microbial Biology, University of California, Berkeley, CA 94720, USA; 3Plant Gene Expression Center, USDA Agricultural Research Service, Albany, CA 94720, USA

**Keywords:** amino acid signaling, target of rapamycin, metabolism, mTOR, Sestrin2, GCN2, Castor1, GATOR, Ragulator, *Arabidopsis thaliana*

## Abstract

Eukaryotic cells monitor and regulate metabolism through the atypical protein kinase target of rapamycin (TOR) regulatory hub. TOR is activated by amino acids in animals and fungi through molecular signaling pathways that have been extensively defined in the past ten years. Very recently, several studies revealed that TOR is also acutely responsive to amino acid metabolism in plants, but the mechanisms of amino acid sensing are not yet established. In this review, we summarize these discoveries, emphasizing the diversity of amino acid sensors in human cells and highlighting pathways that are indirectly sensitive to amino acids, i.e., how TOR monitors changes in amino acid availability without a *bona fide* amino acid sensor. We then discuss the relevance of these model discoveries to plant biology. As plants can synthesize all proteinogenic amino acids from inorganic precursors, we focus on the possibility that TOR senses both organic metabolites and inorganic nutrients. We conclude that an evolutionary perspective on nutrient sensing by TOR benefits both agricultural and biomedical science, contributing to ongoing efforts to generate crops for a sustainable agricultural future.

## 1. Introduction

Target of rapamycin (TOR) is a serine/threonine kinase that senses environmental cues, especially nutrient availability, to coordinate eukaryotic cellular metabolism [1,2,3,4,5]. TOR supports growth by activating anabolic processes, such as mRNA translation [6,7,8,9], nucleotide biosynthesis [10,11,12,13], and lipid biosynthesis [14,15,16], while inhibiting catabolic processes, such as autophagy [17,18,19,20,21]. TOR is especially responsive to amino acid signals. Free amino acids stimulate TOR, which then increases the global rate of protein synthesis to metabolize those amino acids [7,22,23], thereby maintaining metabolic homeostasis while promoting growth and development. The stimulatory effect of amino acids on TOR has primarily been studied in heterotrophs, especially yeast, invertebrates, and mammals, that rely on dietary sources for the 20 proteinogenic amino acids. Unlike these model species, plants are autotrophs that can synthesize all proteinogenic amino acids from inorganic precursors [24,25,26]. TOR is also responsive to amino acid signals in autotrophic plants [27,28,29,30], but the precise amino acids sensed by TOR and the molecular mechanisms of amino acid sensing in plants remain unknown.

In this review, we summarize the current understanding of the molecular pathways of amino acid sensing by TOR in mammalian, invertebrate, and yeast models, discuss the recent literature on amino acid sensing by TOR in plants, address the possibility that TOR responds to both inorganic and organic nitrogenous compounds in plant cells, and propose that a deeper understanding of nitrogen–TOR signaling is urgently needed to minimize reliance on chemical fertilizers for a sustainable agricultural future. We argue that a comparative, evolutionary perspective on nutrient sensing by TOR benefits both agricultural and biomedical science, and we highlight how ongoing studies of TOR signaling in plants and algae contribute to these fields.

## 2. Sensors and Transducers in Metabolic Signaling

Multiple amino acid sensors are proposed to regulate TOR activity in humans [23,31]. The relative significance of these sensors and their precise mechanisms of signal transduction are under debate. Therefore, we now define several terms in the field of metabolic signal transduction that are useful for understanding the amino acid–TOR network. A “sensor” protein responds directly to an environmental cue, e.g., by binding to a metabolite that alters the protein’s activity or by undergoing a structural change triggered by light or temperature. A classical example of a sensor from plant biology is phytochrome, which reversibly changes conformations in response to red or far-red light [32]. Sensors then engage “transducer” proteins, either indirectly or through direct protein–protein interactions, which may either act on additional transducers to create a chain of signaling events in a pathway or activate the response to a cue. Antibodies, ligand-binding proteins, and hormone receptors are examples of sensors [33]; mitogen-activated protein (MAP) kinases, G-proteins, and TOR itself are examples of transducers.

The distinction between “sensor” and “transducer” is crucial for understanding nitrogen- and amino acid responsive signaling networks in eukaryotic cells. Debates over the function of GCN2 (general control nonderepressible 2), for example, revolve around whether GCN2 is a sensor or a transducer. GCN2 is a serine/threonine kinase that is activated when cells experience extreme stress [34]. Among other substrates, GCN2 phosphorylates the eukaryotic translation initiation factor, eIF2α (at serine 51 in human eIF2α, which is orthologous to serine 56 in Arabidopsis eIF2α), which globally represses translation of most mRNAs and selectively upregulates translation reinitiation of transcripts that encode small, upstream open reading frames (uORFs) in addition to the primary ORF [35,36,37,38]. Transcripts with uORFs encode various stress-inducible and starvation-responsive transcription factors [39]. Thus, when GCN2 is active, gene expression is regulated translationally and transcriptionally to promote stress responses.

GCN2 is often misleadingly called an “amino acid sensor” as GCN2 is rapidly activated in response to amino acid deprivation and mediates starvation responses in diverse eukaryotes [39,40]. Early studies confirmed that GCN2 does not directly sense amino acids and hypothesized that GCN2 senses uncharged tRNAs (i.e., tRNAs that have not been ligated with an amino acid by an aminoacyl-tRNA synthetase, presumably due to amino acid deprivation), and that physical interaction with uncharged tRNAs allosterically activates GCN2 [40,41,42,43]. Later studies upended this straightforward model, proposing instead that GCN2 senses translational stress through alternative mechanisms, such as by directly binding the exposed phosphoprotein (P)-stalk of stalled mRNA-bound ribosomes [44,45] or by acting as a transducer downstream of other proteins that associate with stalled ribosomes [46,47]. While the distinction among directly sensing uncharged tRNAs, directly sensing exposed P-stalks, and acting as a transducer for other sensors of translational stress seems esoteric, these distinctions are critical for understanding human diseases [48]. In mouse, *Drosophila*, and human cell models of Charcot-Marie-Tooth (CMT) disease, a genetic disorder that causes peripheral neuropathies, tRNA synthetases carry partially dysfunctional mutations that limit the synthesis of charged tRNAs, leading to translational stress and constitutive activation of GCN2 [48,49,50,51]. Overexpressing tRNA genes reduces CMT symptoms and suppresses GCN2 by increasing levels of both charged tRNAs and uncharged tRNAs [49,51]. This experiment in a disease model demonstrates that accumulation of uncharged tRNAs is not sufficient to activate GCN2 in cells. Therefore, structural, biochemical, and in vivo experiments indicate that GCN2 is not an amino acid sensor or an uncharged tRNA sensor but is most likely a sensor or transducer of translational stress per se in eukaryotic cells [46,47,52].(Figure 1)

## 3. The Growing Chorus of Amino Acid Sensors and Transducers for Mammalian TOR

From an evolutionary perspective, sensors may evolve through one of two general mechanisms: exaptation or adaptation [2]. In the former case, a sensor evolves from a protein that already interacted with the cue it senses (e.g., an enzyme and its substrate), later evolving to interact with a transducer that connects to a downstream signal transduction pathway. This situation is called “exaptation” [53] because both the transduction pathway and the sensor already existed in cells, and they were co-opted to create a new pathway that conferred some fitness benefit. When the sensor retains its ancestral function in addition to its new, exaptive role as a sensor, the sensor is often called a “moonlighting protein”. In the latter case, an existing transducer or sensor evolves the capacity to sense a new cue. This situation is called “adaptation” because the sensor did not already interact with a cue but arose de novo through mutation of a previously nonsensor protein and conferred some fitness benefit. The distinction between exaptation and adaptation is useful for understanding the evolutionary origins and diversity of amino acid sensors in eukaryotic cells.

We now present the major proposed amino acid sensors for TOR and discuss their proposed transducers that mediate the signaling pathway to TOR. In mammals, TOR is especially responsive to levels of the essential amino acid leucine (Leu) and the conditionally essential amino acid arginine (Arg) [54,55]. Leucyl-tRNA synthetase (LARS) was the first proposed leucine sensor for TOR, identified through a protein–protein interaction screen for potential amino acid sensors in yeast [56] and through colocalization experiments in mammalian cells [57]. SLC38A9, an amino acid transporter localized to the membranes of lysosomes (lytic vacuoles), was the first proposed arginine sensor for TOR, identified through a protein–protein interaction screen for potential amino acid sensors in mammals [58]. Both LARS and SLC38A9 are examples of exaptation as their ancestral functions already involved direct interactions with leucine and Arg, respectively, at a molecular level.

Protein–protein interaction screens later identified additional amino acid sensors for TOR in mammalian cells, including the leucine sensors Sestrin1/2 [59,60] and Sar1b [61], the arginine sensor Castor1 [62,63], and a sensor of the methionine derivative S-adenosyl methionine (SAM) called SAMTOR [64]. Sestrins have an ancestral role as transducers of stress signals to TOR [2,65]: *Sestrin* genes are transcriptionally activated by stress-responsive transcription factors, including p53 and the GCN2-stimulated activating transcription factor 4 (ATF4). Very recently in the mammalian lineage, some Sestrins evolved a leucine-binding pocket that induces a conformational change to prevent their role in transducing stress signals to TOR. In humans, not all Sestrin paralogues interact with leucine (for example, Sestrin3 is completely leucine-insensitive), and Sestrin orthologues outside of mammals do not include a leucine-binding site, suggesting that the leucine sensor function of Sestrin1/2 is a recent adaptation and that *Sestrin* genes are subfunctionalizing in the human lineage [65]. Sar1b is a small GTPase in the Arf family, which is conserved across eukaryotes and participates in endomembrane trafficking. Human and *C. elegans* Sar1b orthologues directly bind to leucine and signal leucine levels to TOR through interaction with transducers [61]. Whether Sar1b orthologues in other lineages bind to leucine is not established, but presumably, the role of Sar1b as a leucine sensor is another example of adaptation. The evolutionary histories of Castor1 and SAMTOR are less clear; Castor1 encodes two ACT domains, which are found in various metabolite-binding proteins and enzymes [66], and SAMTOR likely evolved from a SAM-dependent methyltransferase or may itself be a functional SAM-dependent methyltransferase enzyme [64]. Neither gene has readily identifiable orthologues outside of vertebrates, suggesting that these sensors evolved very recently in the human lineage.

The transducers of amino acid signals to TOR were largely identified before the amino acid sensors, primarily through screens for protein interactors of TOR and associated proteins, but also through genetic screens for regulators of TOR activity. The Rag family of small GTPases forms a complex with regulatory proteins called the “Ragulator”, which forms a platform for the activation of TOR at the surface of lysosomes [67]. These Rag GTPases are regulated by the “GAP activity towards the Rags 1” (GATOR1) complex, which stimulates hydrolysis of RagA/RagB-bound GTP to negatively regulate TOR [68,69,70]. GATOR1 is negatively regulated by another multiprotein complex, GATOR2. GATOR2 transduces signals from Sestrins, Castor1, and Sar1b; GATOR1 transduces these signals along with signals from SAMTOR; and the Ragulator complex transduces all these signals along with signals from SLC38A9 and LARS (Figure 2).

Multiple groups proposed that the diversity of amino acid sensors in mammalian cells reflects functional differences rather than simple redundancy [23,61]. The proposed sensors have distinct binding affinities, subcellular localizations, and expression profiles, and sensors intersect with distinct transducers upstream of TOR, which conceivably allows the various sensors to coordinately act on TOR like “rheostats” that fine-tune TOR activity in response to dynamic metabolic conditions [1]. To illustrate, Castor1 and SLC38A9 are localized to the cytosol and lysosome, respectively, responding to different subcellular pools of arginine [58,62,63,71]. As another example, Sestrin1, Sestrin2, and Sar1b each bind specifically to leucine in the cytosol, but with high, moderate, and low affinities, respectively [59,61]. In cells that express both Sestrins and Sar1b, Sestrins engage with GATOR2 and only partially suppress TOR activity when leucine levels are moderately low, but both Sar1b and Sestrins engage with GATOR2 and completely inactivate TOR when leucine is extremely scarce or absent [61].

Beyond leucine and arginine, many other amino acids stimulate TOR, and the potency of TOR activation by specific amino acids varies across biological contexts (e.g., cell type, species, experimental conditions, etc.). Some of these amino acid sensitivities are mediated by the GATOR2–GATOR1–Ragulator transduction pathway, but others act independently. For example, glutamine and asparagine activate TOR through Ragulator-independent mechanisms [72,73]. The details of how glutamine and asparagine activate TOR are still under investigation, but several studies have pointed to the role of another small GTPase, Arf1, in transducing glutamine and asparagine signals to TOR [74]. Alternative (but not mutually exclusive) pathways have been proposed, including that glutamine drives synthesis of α-ketoglutarate (αKG, also known as 2-oxoglutarate or 2OG) via glutaminolysis, and that αKG promotes TOR activity in Ragulator-dependent mechanisms [75]; or, that glutamine and asparagine elevate ATP/AMP ratios via asparagine synthase and the GABA shunt, which, in turn, inhibits the TOR-antagonizing AMP-activated kinase (AMPK) [75]. Therefore, glutamine and asparagine may activate TOR indirectly, without any glutamine- or asparagine-specific sensor proteins in cells.

Leu may induce TOR indirectly, without a *bona fide* Leu sensor, under some circumstances. In cells that do not strongly express Sestrins, including HeLa cells, Leu stimulates TOR through a downstream metabolite, acetyl coenzyme A (AcCoA) [76,77]. AcCoA is synthesized in human cells from pyruvate, fatty acid, or branched-chain amino acid (especially Leu) precursors [78]. Under leucine deprivation, AcCoA levels temporarily decrease, but they can be restored by resupplying leucine or by compensatory synthesis from other precursors [76]. AcCoA is mobilized in the cytosol to acetylate the TOR-associated protein RAPTOR, which effectively increases TOR activity in cells; RAPTOR acetylation correlates tightly with cytosolic AcCoA concentrations, thereby acting as an AcCoA sensor [77]. Knocking down Sestrin1/2 or LARS does not prevent leucine from activating TOR in some cell types (e.g., HeLa cells), but knocking down the enzymes that metabolize leucine to produce AcCoA makes TOR unresponsive to leucine supply [76,77]. This discovery illustrates the complexity of amino acid signaling in cells and highlights how TOR can integrate multiple dynamic cues as the hub of a signal transduction network.

## 4. Plant Nutrient Sensing: From Inorganic Precursors to Organic Metabolites

Most investigations relevant to plant amino acid signaling have focused on how upstream precursors are sensed by plant cells, including inorganic nutrients absorbed from soil (e.g., nitrate, ammonium, and sulfate) and photosynthesis-related cues (carbon dioxide, light, and sugars). The best-studied sensor of nitrogenous nutrients in plant cells is the plasma-membrane-localized nitrate transporter 1.1 (NRT1.1, often called chlorina 1 or CHL1), a transceptor that alters nuclear gene expression when it detects environmental nitrate via a calcium-dependent protein kinase signaling cascade [79,80,81]. Pioneering studies of plant TOR signaling focused on how TOR reacts to photosynthesized sugars rather than amino acids. In plant cells, sugars are directly sensed by proteins including hexokinase (HXK1) and the SNF1-related protein kinase (SnRK1, orthologous to human AMP-activated kinase, AMPK, but not sensitive to AMP or ATP) [82]. HXK1 evolved an exaptive signaling role in addition to its critical metabolic function in glycolysis, signaling sugar availability in response to direct interaction with glucose [83,84,85]. SnRK1 phosphorylates proteins to promote stress and starvation responses but is repressed by a proposed direct interaction with trehalose-6-phosphate, an intermediate in sugar metabolism [86,87,88,89]. Although TOR activity is likely regulated by SnRK1 [90], sugar activation of TOR also requires glycolysis and oxidative phosphorylation [91,92], suggesting that TOR responds primarily to ATP levels rather than directly to sugars. This may be analogous to the mammalian model: TOR senses glycolytic intermediates (dihydroxyacetone phosphate), but only in cells that lack ATP-sensing pathways [93]. Although no ATP sensors are established in plant cells, a proposed ATP sensor for mammalian TOR, the cochaperone R2TP ATPase complex, is conserved in plants and regulates TOR activity, hinting that R2TP may be an ATP sensor in plant cells [94,95].

Several forward genetic screens have refocused attention on how plants sense organic nitrogenous nutrients, especially amino acids and nucleotides. Two independent genetic screens for *Arabidopsis thaliana* mutants that disrupt cellular patterning identified recessive alleles of *isopropyl malate synthase 1* (*IPMS1*), which encodes an enzyme in the leucine biosynthetic pathway [27,28]. These *ipms1* mutants display defects in cytoskeletal organization and leaf shape during early seedling development [28], and *ipms1* resolves the root hair abnormalities observed in *leucine-rich receptor*/*extensin 1* (*lrx1*) mutants [27]. Moreover, *ipms1* mutants exhibit elevated TOR activity, can be partially phenotypically rescued by very low concentrations of TOR inhibitors, and are resistant to the growth-inhibiting effects of moderate concentrations of TOR inhibitors [28]. Therefore, the effects of *ipms1* on amino acid metabolism are somehow transduced to TOR, and this is likely the primary cause of *ipms1* cellular and developmental phenotypes. A functional genetic screen for *Nicotiana benthamiana* genes that regulate TOR activity identified *phosphoribosyl pyrophosphate synthetase 4* (*PRS4*), which encodes an enzyme required for nucleotide biosynthesis in plant cells [13]. Plant TOR senses purine and pyrimidine availability, analogous to how mammalian TOR monitors nucleotide levels [13,20]. Thus, plant TOR reacts to disruptions in organic nutrient levels and biosynthetic pathways, but the sensors and transducers involved remain enigmatic. (Figure 3)

Beyond plants, the intersecting roles of nutrient sensing, amino acid metabolism, and TOR signaling in distantly related microalgae, such as the chlorophyte *Chlamydomonas reinhardtii*, are currently under investigation and have potential industrial applications. Several algae are excellent sources of triacylglycerols (TAGs) that could serve as biofuel feedstocks for a sustainable energy future [96], and inhibiting TOR is sufficient to significantly induce TAG biogenesis in candidate biofuel species, including *C. reinhardtii* and *A. thaliana* and the nongreen algae *Cyanidioschyzon merolae* and *Phaeodactylum tricornutum* [97,98,99,100,101]. Alongside TAG accumulation, inhibiting TOR rapidly increases amino acid levels in *C. reinhardtii* cells [102]. Metabolomic investigations of the origin of these amino acids found that amino acid accumulation upon TOR inhibition in *C. reinhardtii* is not primarily due to suppression of mRNA translation or autophagic recycling of proteins but instead due to drastically elevated inorganic nitrogen uptake from the environment and subsequent de novo amino acid biosynthesis [103]. As TOR activity is stimulated by both nitrogen and carbon sources in *C. reinhardtii* cells [104], a possible model is that TOR monitors intracellular nutrient status to regulate extracellular nutrient uptake and maintain carbon/nitrogen balance [102,103,104]. Understanding how algal cells monitor nutrient status on a molecular scale may accelerate efforts to engineer algae for efficient biofuel production, but very little is known about the mechanisms of nutrient sensing or which nutrients are sensed (inorganic nutrients or organic forms such as amino acids) in these species.

A major outstanding question in plant metabolic signaling is whether plant cells encode true amino acid sensors analogous to the aforementioned mammalian amino acid sensors for TOR. Amino acid profiles of *ipms1* mutants, which show constitutively elevated TOR activity, revealed altered levels of almost every amino acid except for lysine and methionine [28]. The relative potency of TOR activation by amino acids was not clearly resolved by experiments using other mutants in the branched-chain amino acid biosynthetic pathway [28]. In leaf discs floated on solutions containing either isoleucine or glutamine at night, TOR activity was rapidly induced [29]; no amino acids were conclusively shown to be incapable of activating TOR, but glutamine had a stronger effect on TOR activity than isoleucine [29]. In Arabidopsis seedlings grown for 9 days without any source of nitrogen, TOR became inactive and growth was arrested. Supplying most amino acids to these nitrogen-starved seedlings activated TOR within minutes, except for arginine, proline, and the aromatic amino acids phenylalanine, tryptophan, and tyrosine [30]. Nitrate and ammonium may also activate TOR in this experimental system, even in the presence of tungstate (a nitrate reductase inhibitor) or methionine sulfoximine (a glutamine synthetase inhibitor), which prevent the assimilation of nitrogen into amino acids [30]. This may suggest that TOR specifically senses inorganic forms of nitrogen in plants and that amino acids supplied to N-deprived seedlings are catabolized to yield inorganic nitrogen forms. Alternatively, plant TOR may monitor multiple nitrogenous cues, including inorganic nitrogen and diverse amino acids, analogous to the diversity of amino acids and amino acid related metabolites sensed by mammalian TOR. To summarize, although current data do not resolve whether plant cells directly sense amino acid cues, TOR dynamically reacts to changes in amino acid metabolism. (Figure 4)

## 5. The Future of Crops: Deploying TOR for a Sustainable Agricultural Future

Modern agriculture relies heavily on inorganic fertilizers to drive growth and increase yields [105,106]. Fertilizers are primarily composed of ammonias to provide nitrogen (e.g., ammonium nitrate), phosphate rock to provide phosphorus, and potash to provide potassium. Crops use fertilizers inefficiently, resulting in significant fertilizer run-off that is environmentally disruptive [26,107]. Moreover, fertilizers are nonrenewable resources: potash and phosphate rock are both ores mined from limited underground sources. Most studies agree that both potash and phosphate production will peak in the 21st century, reducing fertilizer availability while agriculture faces other pressures from changing climates, dwindling arable land, and growing global populations [108,109]. Therefore, a goal of plant biology in the 21st century is to reduce reliance on external fertilizer sources and maximize nutrient use efficiency in diverse, resilient crop species.

We argue that a detailed mechanistic understanding of the TOR signaling network in plants is critical for these efforts. By defining the nutrient sensors and signaling network transducers that act upstream of TOR and the phosphoprotein effectors and downstream processes engaged by TOR, we expect to discover new targets for breeding, biotechnological interventions, or both to improve plant nutrient use efficiency. Whereas TOR signaling networks evolved to maintain homeostasis for plants that experience unpredictable, fluctuating environments in ecological competition with other plant species, domesticated crops do not face these same fitness costs and selective pressures. Ongoing investigations of TOR signaling in model systems, including Arabidopsis, may illuminate new targets for genetic modification, contributing to the larger project of establishing a sustainable global agricultural program.

## Figures and Tables

**Figure 1 biomolecules-12-00387-f001:**
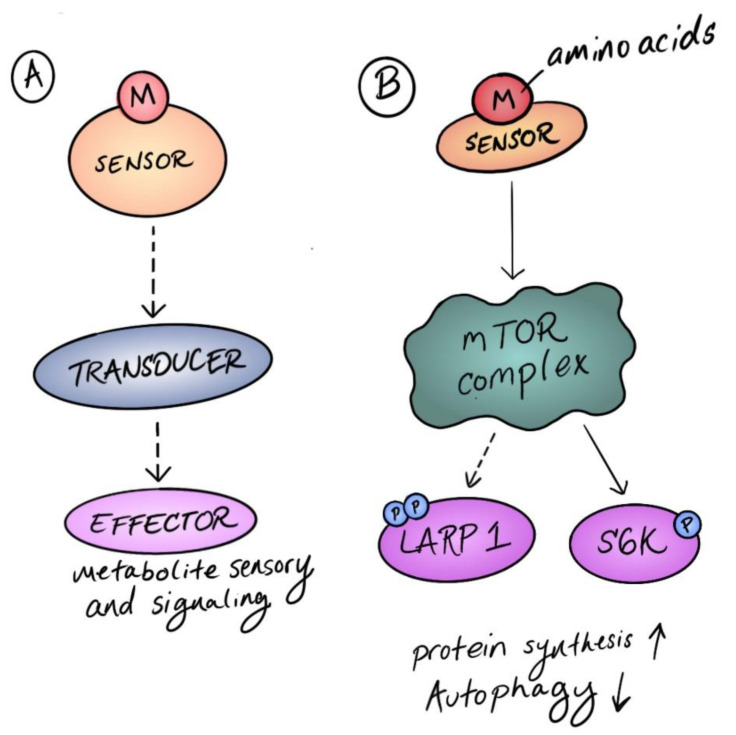
Sensors and transducers in metabolic signaling. (**A**) Metabolic signaling is triggered by a metabolite sensor protein that directly binds to a metabolite. The sensor then engages transducers in a signaling pathway that eventually activate responses through an effector protein. (**B**) In mammalian models, amino acids are sensed directly by diverse sensor proteins that activate the mechanistic target of rapamycin (mTOR), a central regulatory hub that transduces diverse upstream signals to coordinate metabolism. mTOR then phosphorylates additional transducers and effectors, such as LARP1 and S6K, to promote protein synthesis and repress autophagy.

**Figure 2 biomolecules-12-00387-f002:**
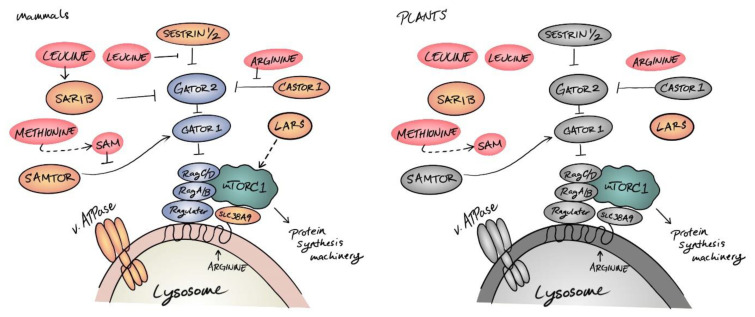
Amino acid signaling in eukaryotes. In mammals, amino acids (red) are monitored by multiple sensors (orange) that engage transducers (blue) at multiple steps in the GATOR2–GATOR1–Ragulator–Rag GTPase signaling cascade to coordinate TOR (teal) activity at the surface of the lysosome. Very few of these signaling components are conserved in plants (gray indicates not conserved in plants). Sar1b and LARS, which have essential roles in membrane trafficking and tRNA synthesis, respectively, were exapted as amino acid sensors in the animal lineage and may not act as amino acid sensors in plants.

**Figure 3 biomolecules-12-00387-f003:**
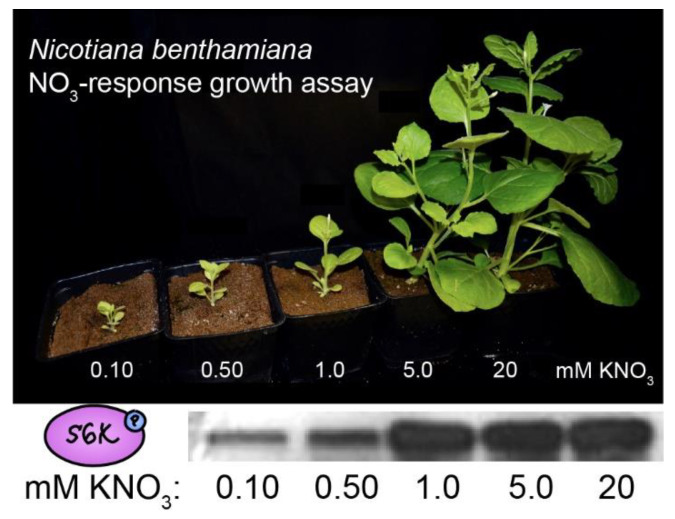
TOR monitors nitrogenous nutrient availability in plants. Plants can synthesize all 20 proteinogenic amino acids from inorganic precursors (nitrate or ammonium, carbon dioxide, and sulfate). *N. benthamiana* were grown on calcined clay and supplied with standard nutrients except for nitrogen, which was supplied as potassium nitrate at the indicated concentrations. Nitrate strongly stimulated plant growth and activated TOR, as measured by Western blots using phosphospecific antibodies against the canonical TOR substrate, S6K-pT449 (methods as in [13]).

**Figure 4 biomolecules-12-00387-f004:**
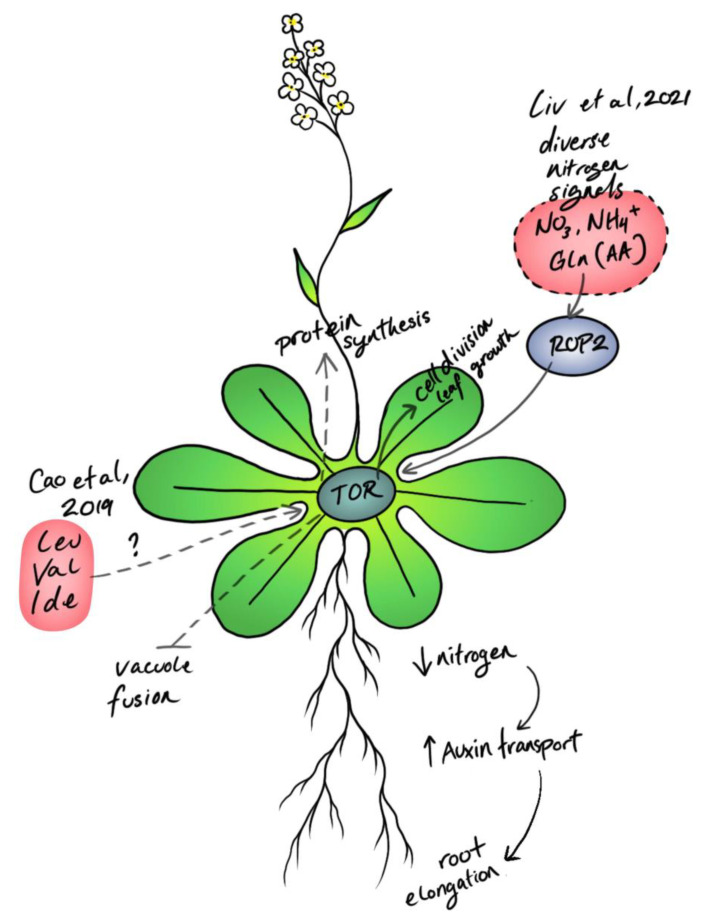
TOR responds to nitrogen and amino acid metabolism in plants to coordinate growth and development [30], protein synthesis [6], and cytoskeletal function [27,28].

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
