# Peer review of "Amino Acid Signaling for TOR in Eukaryotes: Sensors, Transducers, and a Sustainable Agricultural fuTORe"

_biomolecules, 2022, doi:10.3390/biom12030387_

Round 1

Reviewer 1 Report

In this manuscript, the authors review the recent progress in the researches on nitrogen-TOR signaling, illustrate very nice examples to explain the concept difference between “sensor” and “transducer ” that, I agree, is crucial to understand the complex signaling network. They compare the conservation and difference of molecular mechanism of amino acid activated TOR signaling between mammals and plants, and depict a great potential of TOR studies for a sustainable agriculture future. It is a well written and impressive review. I fully endorse publication ASAP of this manuscript. I only have several smaller comments where I suggest the authors to adjust or rephrase.

  1. Line 73, “…that encode small open reading frames (ORFs) upstream of the primary ORF”. This sentence is not clear to me. Is “…that contain small open reading frames (ORFs) upstream of the primary 5’UTR”?
  2. Line 78, “understood” might change to “found”
  3. Line 208, “inorganic soil nutrients” might change to “inorganic nutrients absorbed from the soil”
  4. 215-221, I would suggest to state HXK1 first, then SnRK1, since HXK1 is a more acceptable sugar direct sensor, while SnRK1 senses trehalose-6-phophate, an intermediate in sugar metabolism.
  5. Line 255-256, “the relative potency of amino acid activation by amino acids…” should be “the relative potency of TOR activation by amino acids”.

Author Response

Thank you for the supportive review and thorough reading of the manuscript, we are very glad that you caught these sentences with errors and/or ambiguities!

All points have been addressed in the revised manuscript.

Reviewer 2 Report

The manuscript contain interesting information concerning the TOR role and function in plants. I consider that this information contribute to enrich the analysis of this important kinase and how it could work to control plant growth.

However, the authors must check the comprenhension and the connection between paragraphs.

I consider that the manuscript is mostly well written, however, there is a paragraph that to me is not easy to understand, for example, the authors describe the “function of GCN2 (General Control Non-derepressible 2), for example, revolve around whether GCN2 is a “sensor” or a “transducer” but they do not connect its putative role with TOR. The authors end this paragraph doing a revision of GCN2 but I don’t understand, again, its relation as a possible sensor, direct or indirectly, of amino acids and TOR activity. I think that is the main paragraph confuse to me.

Author Response

Thank you for the supportive review.

We made several revisions to improve style (highlighted in the revised manuscript), including those also suggested by other reviewers.

Although there is some evidence linking TOR and GCN2, this relationship is complicated.  In an earlier draft, we had actually included an entire extra paragraph on this topic, but when we asked colleagues to read the draft, they unianimously agreed that this paragraph detracted focus from the paper, so we removed it.  Our goal is to use GCN2 as an illustrative example of how a signaling pathway might react to changes in amino acid levels without actually functioning as an amino acid sensor.  To avoid making the paper too complicated, we leave discussion of the details of GCN2 function and the evolution of its sensing mechanisms in plants to experts in that field, and here only focus on using it to show the difference between "sensor" and "transducer".

Reviewer 3 Report

Review by Lutt and Brunkard is nicely written and very well explain about the amino acid signalling for TOR in eukaryotes with special reference to it application in sustainable agriculture. The article is accepted for publications in the special issue. However, I have one minor suggestions

  1. As authors mentioned in the MS that nutrient sensing by TOR is very important for agriculture and algal biomass production. It is good if authors will write one short section/para on nutrient sensing by TOR in microalgae and its application and future perspective.

Author Response

Many thanks for the supportive review and excellent suggestion.

We added a paragraph on nutrient sensing, amino acid metabolism, and TOR in green and non-green algae to the review, which we agree enhances the review and expands the potential pool of readers.  so, thanks!